# Quantum Circuit Components for Cognitive Decision-Making

**DOI:** 10.3390/e25040548

**Published:** 2023-03-23

**Authors:** Dominic Widdows, Jyoti Rani, Emmanuel M. Pothos

**Affiliations:** 1IonQ, Inc., 4505 Campus Drive, College Park, MD 20740, USA; 2College of Engineering, University of California, Berkeley, CA 94720, USA; 3Department of Psychology, City, University of London, London EC1V 0HB, UK

**Keywords:** cognitive decision-making, quantum cognition, quantum computing

## Abstract

This paper demonstrates that some non-classical models of human decision-making can be run successfully as circuits on quantum computers. Since the 1960s, many observed cognitive behaviors have been shown to violate rules based on classical probability and set theory. For example, the order in which questions are posed in a survey affects whether participants answer ‘yes’ or ‘no’, so the population that answers ‘yes’ to both questions cannot be modeled as the intersection of two fixed sets. It can, however, be modeled as a sequence of projections carried out in different orders. This and other examples have been described successfully using quantum probability, which relies on comparing angles between subspaces rather than volumes between subsets. Now in the early 2020s, quantum computers have reached the point where some of these quantum cognitive models can be implemented and investigated on quantum hardware, by representing the mental states in qubit registers, and the cognitive operations and decisions using different gates and measurements. This paper develops such quantum circuit representations for quantum cognitive models, focusing particularly on modeling order effects and decision-making under uncertainty. The claim is not that the human brain uses qubits and quantum circuits explicitly (just like the use of Boolean set theory does not require the brain to be using classical bits), but that the mathematics shared between quantum cognition and quantum computing motivates the exploration of quantum computers for cognition modeling. Key quantum properties include superposition, entanglement, and collapse, as these mathematical elements provide a common language between cognitive models, quantum hardware, and circuit implementations.

## 1. Introduction

Human behavior often evades predictions made by mathematical laws, and models based on classical mechanics and probability have sometimes proved disappointing [1]. Much of the information we depend on is open to doubt, almost all of the predictions we make using this information are uncertain [2], and even models based on the assumption that the future will be like the past, even in a statistical sense, have sometimes been quite inadequate [3] (Ch. 8). Events and beliefs can be connected in ways we do not always understand, and we cannot always explore such situations without disturbing them.

In light of these problems, some traditional objections to quantum theory look like virtues. Unpredictability is part of the universe. Sometimes observing one thing makes another harder to observe. Asking a question can itself force a decision that closes other options. Once someone has chosen a position, even arbitrarily, they are likely to stick to it unless the situation evolves.

Motivating analogies alone are not evidence that quantum theory or computing are better at modeling or implementing human-like behavior. Such evidence has been carefully collected in works including those of Blutner et al. [1], Aerts [4], Busemeyer and Bruza [5]. The correspondences are mathematical rather than physical, involving vectors, bases, matrix operations, and projections, rather than waves, particles, spin, and angular momentum. Quantum cognitive models do not depend on their implementation using quantum mechanics at the atomic level, any more than the use of calculus in classical economics depends on the ability to trade infinitesimal amounts of money.

Now in 2023, we have working quantum computers as well. These machines are at an early rapid-growth stage: for example, the machine used in the experiments reported here uses 11 qubits [6], and the number of available and reliable qubits is already in the twenties, which can support larger and deeper circuits [7]. The time is ripe to investigate what these machines are capable of, especially in fields where quantum mathematical concepts have already proved useful. Such approaches are being tried in natural language processing [8,9] and economics [10].

In this paper, we show some initial quantum circuit implementations that tie the cognitive and mathematical modeling together into intuitive building-blocks. The models are taken from explicit small-scale examples in the psychology literature, so the implementations described can all be comfortably simulated on a modest classical computer—the results here do not claim to show ’quantum advantage’ in a computational sense. Instead, we show that very small circuits—some as small as a single qubit—can model surprisingly human scenarios. In addition, it is sometimes easy to compose or wrap these components: for example, using entanglement with one extra qubit can activate different behaviors for different inputs.

The paper proceeds by reviewing some long-established violations of classical set-theoretic reasoning (Section 2), and successful alternatives based on vectors rather than sets developed by quantum cognition researchers (Section 3). Section 4 introduces quantum circuits, with enough detail to help understand the quantum circuits for order and disjunction effects which are developed in Section 5, Section 6 and Section 7. Section 8 describes the implementation and results on quantum hardware, and Section 9 discusses related and further work.

## 2. Human Violations of Classical Probability

Everyday human judgments and choices sometimes appear to defy classical probability rules. For example, the order in which we ask questions matters, in ways that violate the classical notion that a conjunction is modeled by an intersection of fixed sets. We begin by explaining this using a well-known example, the Clinton–Gore order experiment described by Moore [11].

In the 1990s, Bill Clinton was the President of the United States, and Al Gore was the Vice President. Al Gore was perceived in public opinion to be “more trustworthy” than Bill Clinton (see below), but the perceptions of trustworthiness varied depending on whether the two people were considered separately or together. This has been discussed from a quantum point of view by several authors including Wang and Busemeyer [12] who describe the results as follows:

In a Gallup poll conducted on 6–7 September 1997, half of the 1002 respondents were asked the following pair of questions: “Do you generally think Bill Clinton is honest and trustworthy?” and subsequently, the same question about Al Gore. The other half of the respondents answered exactly the same questions but in the opposite order.… The results of the poll exhibited a striking order effect. In the non-comparative context, Clinton received a 50% agreement rate and Gore received 68%, which shows a large gap of 18%. However, in the comparative context, the agreement rate for Clinton increased to 57% while for Gore, it decreased to 60%.

Here, the non-comparative context means asking “Is X trustworthy?” on its own, whereas the comparative context means asking “Is Y trustworthy?” first, followed by “Is X trustworthy?”

These findings are not surprising in day-to-day life, and, indeed, they are entirely consistent with social psychology theory (e.g., [13]). We expect that the order in which information is presented affects the way it is perceived—for example, if you had decided to ask for a pay-rise, you would give your manager good news rather than bad news just beforehand. Perhaps more surprising is the fact that order effects *are not* part of classical logic or probability, unless we add new rules that introduce different conditions being applied for observations made in different orders. It is important to understand why this is the case.

The change in outcomes, depending on which order the questions are asked, violates the classical assumption that probability is a measure of the relative sizes of different fixed sets. For example, it would appear from the experiment quoted above that the size of the set *C* of people who believe that “Clinton is honest” changes if people are asked “Do you believe that Gore is honest?” just beforehand. Classical set theory models the answers to the Clinton–Gore questions by assuming that there is a set *P* corresponding to the total population, a subset C⊆P of people who believe that Clinton is honest, a subset G⊆P who believe that Gore is honest, and then the subset who believe that both are honest is given by the intersection C∩G. Whether a participant belongs to one of these sets is unaffected by asking if the participant also belongs to another. Asking a question may cause a system to output an answer, but it does not change the system’s internal state. To account for the changing probabilities observed above, we would need to be able to model the question “Is *x* in *G*?” as partly depending on whether the model has been asked “Is *x* in *C*?”.

There are several other known examples that violate the basic classical assumption that probability judgments depend on comparing the relative sizes of fixed sets [14]. These include so-called conjunction fallacies and disjunction effects, where the probability of a combination of events is considered to be greater or less than what would be possible if the probability for each event is measured and the probabilities are then combined [15].

A well-known example is the Prisoner’s Dilemma paradox, in which there are two prisoners with no means of communication, whom the police have reason to believe may be connected to the same crime. Each prisoner is offered two choices:‘Betray’ and attest that the other prisoner is a partner in crime, or,‘Cooperate’ by *not* implicating the other prisoner in the crime.

If both prisoners cooperate, they both go to jail for 2 years. If one cooperates and the other betrays, then the betrayer only goes to jail for 1 year, and the cooperator for 5 years. If both betray one another, both go to jail for 3 years. There are many variants of this setup, studied in psychology and game theory since at least the 1950s [10] (§8.2).

From the way the game is setup, even though the smallest jail time overall is when both partners cooperate, each participant gets a shorter jail sentence if they betray their partner, whichever choice the partner makes. This cold logic is confirmed in many experiments: when told that their partner has betrayed them, 82% of participants betray, and when told that their partner has cooperated, 72% of participants betray. The paradoxical finding, however, is that the probability of defecting is reduced to 64% if the partner’s choice is unknown. (These are averages across many experiments, summarized by (Busemeyer and Bruza [5], Table 9.4).) This is paradoxical because it violates the “Sure Thing Principle” of Savage [16], or in mathematical terms, the classical law of total probability, which in this case would state that:(1)$$P\left(B\right)=P\left(B|C\right)P\left(C\right)+P\left(B|C^{\prime }\right)P\left(C^{\prime }\right).$$

Substituting in the observed numbers above, if P(B)=64% is smaller than both P(B|C)=82% and $P(B|C^{\prime })=72\%$, and $P\left(C\right)+P\left(C^{\prime }\right)=1$, there is no possible value of P(C) and $P(C^{\prime })$ that could satisfy this equation.

These are two of the many examples from the seminal works of Dan Kahneman and Amos Tversky, amongst others, which have illustrated the systematic bias in judgments, relative to the classical expectation that probabilities should be solely derived from frequencies [14,17,18]. Other well-known examples include conjunction fallacies: for example, participants told that a character, Bill, is intelligent and strong in math, but unimaginative, considered it more likely that “Bill is an accountant who plays jazz for a hobby” than just “Bill plays jazz for a hobby” [14].

Sometimes, violations of classical probability laws have common explanations. In the Prisoner’s Dilemma, participants could be described as giving benefit of doubt to their partners: in some situations, not knowing an outcome may encourage someone to view a situation more positively. Sometimes the opposite happens, for example when we experience risk aversion. Risk aversion is well-known in economics, and has been demonstrated in experiments: for example, in a paper called *The uncertainty effect: When a risky prospect is valued less than its worse outcome*, Gneezy et al. [19] reported that:

For example, people are willing to pay an average of $26 for a $50 gift certificate, but only $16 for a lottery that pays either a $50 or $100 gift certificate, with equal probability.

The analysis of probabilities and the inequalities that must hold between them figures prominently in George Boole’s *The Laws of Thought* [20] (Ch. 16–21) and is developed in Boole’s subsequent works, in which inequalities between probabilities are described as ‘conditions of possible experience’. Boole focused particularly on the issue of compatibility between individual (marginal) and joint probabilities: for example, if 6 out of 10 objects are green and 7 out of 10 are square, there must be at least 6+7−10=3 objects that are both green and square. With this background, one way to consider the work described here is as an extension of Boole’s work on joint and marginal probabilities, to include conditional probabilities of unknown events. This topic is already known to be connected with quantum theory, because the famous Bell and CHSH inequalities of quantum physics are examples of Boole’s conditions, as recognized and explained by Pitowsky [21].

## 3. Quantum Probability Models for Cognitive Behavior

This section describes some examples of the quantum approaches that have been used to model some of the above scenarios more effectively, focusing particularly on order and disjunction effects. The field of quantum cognition has developed in cognitive science since the 1990s, and uses mathematical structures from quantum theory as models for human behavior in ways that solve precisely the problems posed in the previous section. Several authors have contributed key insights and results in this field, which is now established enough to have comprehensive surveys including those of Blutner et al. [1], Pothos and Busemeyer [2], Busemeyer and Bruza [5], and in economics, Orrell [10]. There are many variants of the approaches introduced, and this section is intended to summarize the main themes rather than to provide an exhaustive review. Here we explain key aspects of the models for order and disjunction effects that motivate the subsequent quantum circuit implementations.

Common themes used in these approaches include:The states of systems are represented by vectors.Outcomes of ‘measurements’ (for example, answers to questions) are modeled by projecting state vectors onto eigenvectors representing different outcomes.The probability of a particular choice or answer is determined by the square of the scalar product (hence the angle) between the system’s state and the outcome state.A system can be in a superposition of various different states.Contributions to these superpositions can interfere (constructively or destructively). This sometimes represents cognitive couplings between events.Measuring the system forces it to ‘choose’ or collapse into a specific output state.

A core notion in quantum cognitive models is that the violation of classical probabilities in disjunction effects happens because the states representing the as-yet-unknown outcomes of the events interfere with one another. Then, when the uncertainty is resolved (by telling a subject a particular outcome, or sometimes by changing the wording to encourage this kind of analysis), the scenario ‘collapses’, the available options are reduced, and the interference vanishes.

### 3.1. Example with Order Effects

Order effects such as the Clinton–Gore example have been described using quantum probability as an alternative to classical probability. Quantum probability depends on comparing angles rather than volumes, and crucially, measuring a system causes it to ‘collapse’ from a superposition of states. The state is projected onto whichever pure state is observed, with a probability determined by the magnitude-squared of the projection output. As projections do not commute with one another, the order of projections matters: so the probability of different outcomes depends on the order of measurement. This property has been used successfully to model order effects, including work established in the survey of Busemeyer et al. [22].

These models are often described using images such as those in Figure 1 (for example, see [2], p. 757). It is clear that the projection of the |0〉 vector onto the *Clinton* axis is further from the origin (hence represents a larger probability) if the projection chain goes via the *Gore* (right) axis than when it jumps all the way to *Clinton* at once (left). This follows from the double-angle formula cos(a+b)=cos(a)cos(b)−sin(a)sin(b) with $0\leq a,b\leq \frac{\pi }{2}$, because 0≤sin(a),sin(b), and so cos(a+b)≥cos(a)cos(b).

Key quantum concepts used here are superposition and projection. The initial |0〉 state can be written as a superposition of states representing the different outcomes: for example, a weighted sum of the states representing the beliefs “Clinton is honest” and “Clinton is not honest”. In the process of answering a question, the state vector of the system is projected or ’collapses’ to the state representing the given answer. This is a crucial feature of quantum cognitive models: mathematically, the use of chains of projections that change the state of the system naturally gives rise to order effects, because projection operators do not commute with one another.

### 3.2. Example with Disjunction Effects

Quantum cognitive models use interference to model disjunction effects. The violation of classical probabilities happens because the states representing the as-yet-unknown outcomes of the events interfere with one another, including interference between incompatible outcomes. This can raise or lower the probability of an event, sometimes beyond the bounds that would be possible under the classical law of total probability (Equation (Equation 1)). For example, in the Prisoner’s Dilemma situation, participants appear to demonstrate a ‘benefit of doubt’ tendency: when the first participant’s decision is unknown, the second participant’s probability of defecting is lower. Then, when the uncertainty is resolved (by telling a participant a particular outcome, or sometimes by changing the wording to encourage this kind of analysis), the scenario ‘collapses’, the available options are reduced, and the interference vanishes. Busemeyer and Bruza [5] comment on this as follows:

If choice is based on reasons, then the unknown condition has two good reasons. Somehow these two good reasons cancel out to produce no reason at all! [22] (p. 267).

Similar famous problems introduced by Tversky and Shafir [15] include the Vacation Problem (participants are more likely to book a vacation if the result of an important exam is known rather than unknown, whether the known result is pass or fail) and the Two-Stage Gambling problem (participants are more likely to gamble a second time if they know the outcome of a first gamble, whether or not they won or lost).

The mathematics of such models is described in detail by Blutner et al. [1] (§4) and Orrell [10] (Ch. 4), and for a summary see Pothos and Busemeyer [2] (p. 759). A key part is that the unknown situation “*A* may or may not cooperate” is written as a disjunction $A\vee A^{\prime }$, but this is a quantum rather than a classical disjunction. This means that states of the form $\psi =\lambda A+\mu A^{\prime }$ are contained in $A\vee A^{\prime }$ even if ψ is contained in neither *A* nor $A^{\prime }$ directly. This is called the *non-distributive* property of quantum logic and it explains much of the difference between classical (Boolean) and quantum (vector) logic [23]. This leads to a more general quantum version of the law of total probability which takes the form:$$P\left(B\right)=P\left(B|A\right)P\left(A\right)+P\left(B|A^{\prime }\right)P\left(A^{\prime }\right)+\partial \left(A,B\right)$$
where $\partial \left(A,B\right)=P\left(ABA^{\prime }+A^{\prime }BA\right)$. This term is related to a phase factor φ introduced in the derived equation $P_{v}\left(ABA^{\prime }+A^{\prime }BA\right)=2\sqrt{P_{v}\left(B|A\right)P_{v}\left(A\right)}\sqrt{P_{v}\left(B|A^{\prime }\right)P_{v}\left(A^{\prime }\right)}\cdot \cos\left(\varphi \right)$, where φ relates to the impact of knowing *A* or $A^{\prime }$ for assessing the likelihood of *B*.

An appropriate phase angle φ can be deduced by knowing P(B|A), $P(B|A^{\prime })$ and P(B) when the outcome of *A* is unknown. As the number of terms increases, there are many more potential correlations, and the number of possible interference terms grows exponentially. This is an opportunity and a challenge: we can represent rich and varied correlations, but the number of parameters becomes classically unscalable and there is lots of potential redundancy. Heuristic methods for selecting these parameters have been proposed in the context of quantum Bayesian networks [24].

### 3.3. Bayesian Theory and the Cognitive Relevance of Classical Logic

The departure from classical logic and probability in this paper is not novel. Classical logic as such is no longer considered a viable approach to human behavior, and this was demonstrated convincingly, before quantum cognition started to take shape in the 1990s. It has been generally accepted since the 1960s that human judgments violate classical logical rules, notably since the work of Wason [25]. Wason’s experiments on reasoning demonstrated systematic discrepancies, between conclusions drawn by participants, and the “correct” inference as predicted by the classical “if P then Q” material implication. Psychologists have explored a wide variety of frameworks for alternative theory and models, including heuristics and biases [26], and neural networks (as an example application, see Kurtz [27]).

However, for probabilistic reasoning specifically, an influential approach has been Bayesian probability theory. While it is easily recognized that baseline Bayesian theory cannot accommodate the range of relevant behavioral findings, psychology researchers have sought bounded-rational versions of Bayesian theory [28,29]. The critical point is that the algebraic structure of Bayesian theory is exactly that of classical logic. Indeed, many of the apparent fallacies in probabilistic reasoning are so surprising, because they break seemingly obvious logical constraints (e.g., the conjunction fallacy, as explained in [2]).

While classical logic is no longer employed directly in cognitive modeling, its relevance is still current, because of the enduring interest in Bayesian cognitive models. An analogous point applies to classical circuits: classical circuits acquire ‘relevance’ in current theoretical discussions, exactly because they are the most direct way to implement Bayesian cognitive models. The work in this paper suggests that quantum circuits can play a similarly useful role in the implementation of quantum cognitive models.

### 3.4. Quantum Cognition and Physics

Quantum cognition, so far, has not depended on any physical apparatus such as a human brain being explicitly ‘quantum mechanical’ in its implementation. This point is sometimes confusing, especially as such claims *have* sometimes been made in what might be called ’quantum consciousness’ (see Penrose [30]; a survey of these approaches and their differences is given by Atmanspacher [31]). To the extent that we continue to assume that the human brain is classical (consistently, with much existing evidence, e.g., [32]), then such models are more oriented towards artificial intelligence rather than human cognition, though some recent findings also offer some indications that quantum entanglement is a factor in human consciousness [33]. Whether this distinction is meaningful or not will depend on how much the assumed calculations require quantum physical information. As argued by Orrell [10], this concern is habitually avoided in classical mathematics and mechanics—it is normal to use calculus in economics without worrying about whether the use of mathematics invented for ballistics or planetary dynamics is valid in economics, or whether the force of gravity really applies—and quantum cognition has taken the same pragmatic liberty with quantum mechanics. This approach has made sense partly because so much of the mathematics used in quantum theory has applications elsewhere, including throughout contemporary artificial intelligence [23].

However, another reason for keeping quantum cognition separate from quantum physics, for the first decades of the relevant research, has been that using quantum mechanics itself to implement operations was not a practical option. This is now changing rapidly: the availability of quantum computers has made it possible to implement at least some models of the assumed cognitive operations on quantum hardware. We can now explore this opportunity.

## 4. Quantum Computing and Quantum Circuits Introduction

Quantum computers have become a practical reality, and while still small and noisy, they have been used to perform basic tasks in parts of AI including natural language processing [9,34] and image classification [35]. These works have not demonstrated ‘quantum advantage’ in the sense of performing useful tasks that would be computationally intractable on classical hardware. However, they have demonstrated several proofs-of-concept, with good mathematical reasons for pursuing these further (these include the exponential dimension growth of tensor products, and correspondences between quantum circuits and other structures based on category theory).

### 4.1. Why Bring Quantum Cognition and Quantum Computing Together?

As the opportunities for using quantum computers grow, quantum cognition is a natural place to look for applications, because it already uses common mathematical structures, and addresses areas known to be difficult for classical logic and probability.

Both quantum cognition and quantum computing are constrained by the computational/probability rules in quantum theory. Existing quantum cognitive models are intended to be at the ‘computational level’ of explanation, using the framework of Marr [36]. The computational level concerns the what and the why questions for the system that is studied, that is “what is the goal of the computation, why is it appropriate, and what is the logic of the strategy by which it can be carried out?” [36] (p. 25). Notably, quantum cognitive models address the question of the computational principles which appear to guide behavior.

Quantum computing and circuits offer insight into how quantum cognitive models could be implemented. In Marr’s terms, they concern the algorithmic level of explanation, which concerns process explanations of the studied system, specifically the representations that are employed by the system and the algorithms that operate on the representations to produce an output from an input.

With these thoughts in mind, a quantum circuits implementation of a quantum cognitive model has two main possible aims. First, it could serve as an algorithmic/process proposal for a corresponding quantum cognitive model. Assuming that there are no real quantum mechanical processes in the brain, a putative brain quantum circuit would be epiphenomenal. Second, a quantum vs. classical circuit implementation (of a quantum cognitive model) bears on the question of human vs. artificial intelligence capacity and limits.

Both these aims would require extensive further work to substantiate—the present work provides the foundation for such work, by offering the first principled proposal for quantum circuits corresponding to a quantum cognitive model; and proof of concept, in the form of the simulations conducted directly on a quantum computer.

A related reason for applying quantum computers to quantum cognition is the obvious motivation-by-opportunity—“because we can”. A great deal of investment has gone into developing quantum computers, with anticipated application areas including materials science (e.g., molecular simulation) and logistics (because of combinatoric complexity). It is to be expected that researchers will try to apply these machines to other areas, just as GPUs (Graphics Processing Units) have found extensive applications in machine learning, as well as computer graphics. We expect that, during the 2020s, quantum computers will be tried in many more application areas. Domains with established techniques that already use quantum mathematical models are naturally a promising place to start.

### 4.2. A Brief Primer on Qubits, Quantum Gates and Quantum Circuits

This section explains some of the key aspects of quantum computing needed to understand basic quantum circuits. The proposal of using a quantum mechanical computer to simulate natural processes was made by Richard Feynman in 1982, and by the end of the 1990s, a clear idea of a quantum computer based on quantum bits (qubits) and quantum logic gates had emerged [37] (Ch. 1). In the early 2020s, most quantum programming still involves assembling qubits and gates explicitly, so this introductory section on quantum circuits is like explaining to a reader already familiar with Boolean logic how basic classical logic gates are represented and connected to program a digital computer.

We begin by describing individual qubits. A (classical) bit is a two-level classical system, meaning it can be in one of two states {0,1}. Bits can be represented using different voltages, currents, light intensities, or magnetic polarities, though this material implementation is usually immaterial to the mathematical description. A qubit, by contrast, is a two-level *quantum* system, meaning it can be in a linear superposition of the two states: if the two levels of the system are written |0〉 and |1〉, a qubit can be in a more general state α|0〉+β|1〉. This gives rise to the popular description that a qubit can be in a state ‘between 0 and 1’, but there is more to a qubit than that. A crucial feature of quantum mechanics is that complex numbers are used throughout, for reasons including the fact that the function $f\left(t\right)=e^{it}$ conserves magnitude for real inputs *t*, whereas either the real part cos(t) or the imaginary part isin(t) would lead to fluctuating overall probabilities. Thus, α and β are complex numbers, so the space of possible values α|0〉+β|1〉 has 2+2=4 real dimensions.

The probabilities of observing a qubit in state α|0〉+β|1〉 to be in the states |0〉 or |1〉 are given by ${\left|\alpha \right|}^{2}$ and ${\left|\beta \right|}^{2}$ respectively. It follows that ${\left|\alpha \right|}^{2}+{\left|\beta \right|}^{2}=1$, so the vector α|0〉+β|1〉 has unit length. It also follows that the whole state can be multiplied by any unit complex number $e^{i\theta }$ and give the same measurement probabilities (because $|e^{i\theta }z|=|z|$ for any complex number *z* and real angle θ). These two conditions reduce the number of meaningful parameters needed to describe the state of a qubit from 4 to 2, and this 2-dimensional state-space is called the *Bloch sphere* [37] (§1.2). A standard diagram of this structure is shown in Figure 2. The possible transformation operators on a single qubit—called single-qubit gates—are thus isomorphic to the possible rotations of a 2-dimensional sphere, which has 3-dimensions (generated for example by rotations around the *x*, *y*, and *z*-axes). Thus, a quantum bit is not just more expressive than a classical bit: it is more expressive than is suggested just by saying that it can be ’between 0 and 1’.

The potential of quantum computing becomes even more apparent when we consider multiple qubits. A system with *n* qubits might be measured in any of $2^{n}$ states (all the variants of |1〉|0〉…1, etc., which would be written |10…1〉). Each of these possible states can have its own ’amplitude’ or complex coordinate, so the number of coordinates needed to describe the system is $2^{n}$. Thus, every time we add a qubit, we *double* the number of amplitudes the system can ’hold in memory’ (the quotes being a reminder that we cannot observe these coordinates directly, we can only measure different combinations of zeros and ones). Two qubits are said to be *entangled* when measuring one of them tells us what to expect when measuring the other: for example, with a system in the state $\frac{\sqrt{2}}{2}\left(|00\rangle +|11\rangle \right)$, if its first qubit is measured and found to be in the |1〉 state, it follows that its second qubit will also be measured in the |1〉 state. Two-qubit gates are vital in quantum computing for bringing about such entanglement. One of the most commonly-used two-qubit gates is the Controlled-NOT, or CNOT gate, which performs a complete *X*-rotation (mapping |0〉 to |1〉 and vice versa) on the target qubit if the control qubit is in the |1〉 state. Other gates can be constructed out of these ingredients: for example, a collection of 3 CNOT gates can be used to make a swap gate that swaps the state of two qubits.

A *quantum circuit* is a collection of qubits and gate operations defined in a particular sequence, and typically a *quantum job* is a computation that proceeds by running a quantum circuit several times (each individual run is sometimes called a *shot*), measuring the results from each run, and combining these into a distribution of results that is returned to the user. Figure 3 shows how the gates and operations used in this paper are depicted in standard quantum circuit diagrams.

The quantum circuits that will be introduced in the next sections, for implementing quantum cognitive models including order and disjunction effects, use standard quantum computing circuit-construction methods: there are no new gates or operations needed that make quantum cognition circuits different from quantum computing circuits in general. However, there are limitations in the current generation of quantum computers (including the lack of mid-circuit measurement), which make some implementation choices more convenient than others today. These decisions are discussed in subsequent sections, as the situations arise.

This is a very brief introduction. Readers who are new to this topic are referred to texts such as those of Busemeyer and Bruza [5] (Ch. 10), Orrell [10] (Ch. 5) (specifically focused on quantum cognition), and Bernhardt [38] (Ch. 7), Nielsen and Chuang [37] (Ch. 4) (for quantum computing in general). There are also several good online introductions to quantum gates and circuits. (See, e.g., https://en.wikipedia.org/wiki/Quantum_logic_gate, accessed on March 14 2023 and https://qiskit.org/learn/, accessed on 14 March 2023 c.f. [39].)

## 5. Quantum Circuits for Order Effects

Having introduced quantum cognitive models and quantum circuits, this section starts to bring these together by introducing a quantum circuit implementation for order effects. For this, we return to the example of participants being asked about Clinton or Gore’s honesty in different orders. Each question “Is *X* honest?” is a ‘yes/no’ question, so can be modeled with a 2-state quantum system, i.e., a single qubit. (This generalizes the point in Section 3 that the |0〉 state can be written as a superposition of states representing “Clinton is honest” and “Clinton is dishonest”. The |0〉 state can be written as such a sum for any pair of orthogonal vectors *X* and $X^{\prime }$, and any such *X* can be written as a superposition of |0〉 and ${|0\rangle }^{\prime }=|1\rangle$.)

Thus, a quantum circuit for either the Clinton or the Gore question can be created with a single qubit and a single gate: to model one event with two outcomes (say *A* and not $A=A^{\prime }$), a qubit is assigned to that event, and a single-qubit rotation is applied to set the appropriate output probability. In Figure 4, an *X*-rotation is used, and the angle θ is given by the formula $\cos^{2}\left(\theta \right)=P\left(A\right)⟹\theta =\arccos\left(\sqrt{P(A)}\right)$.

Since both questions are asked in the same scenario, the initial state for both the Gore and Clinton questions can be modeled by a single qubit, while the two questions are represented by measurements along two different axes. With such a setup, we can find the appropriate angles $\theta_{C}$ and $\theta_{G}$ to model the Clinton and Gore probabilities separately, and if $\theta_{G}<\theta_{C}<\pi /2$, we expect that $\cos^{2}\left(\theta_{G}\right)\cos^{2}\left(\theta_{C}-\theta_{G}\right)<\cos^{2}\left(\theta_{C}\right)$, which means that the system is more likely to end up in the *C* state if it is measured in the *G* state as a stepping-stone.

The 2-dimensional nature of the Bloch sphere arising from the use of complex numbers for coordinates adds a useful degree of freedom here, as seen by comparing Figure 1 and Figure 5. In Figure 1, each unit vector is described by a single angle, so the whole system comprising of the initial vector, the Clinton vector, and the Gore vector, is described by just the two angles $\theta_{C}$ and $\theta_{G}$. If P(C)=0.50 and P(G)=0.68, this guarantees that $\theta_{C}\approx 0.785$ and $\theta_{G}\approx 0.601$, so if the angle between these two is guaranteed to be $\theta_{C}-\theta_{G}$, we would infer that $P\left(G|C\right)=\cos^{2}\left(\theta_{G}\right)\cos^{2}\left(\theta_{C}\right)+\sin^{2}\left(\theta_{G}\right)\sin^{2}\left(\theta_{C}\right)\approx 0.668$, which is larger than the value of 57% found in experiments. In this one-dimensional model, Clinton’s perceived trustworthiness gets too much of a boost by being associated with Gore’s.

In the 2-dimensional Bloch sphere model of Figure 5, instead of being predicted by a single angle θ, the Clinton and Gore vectors also have a phase angle φ, and an appropriate combination of rotations can be used to generate any of these states. So as well as fitting the $\theta_{C}$ and $\theta_{G}$ parameters to give the expected probabilities for *C* and *G* on their own, we can also find a parameter φ that fits the expected probability of transitioning from *G* to *C*, as in Figure 5. This enables the model to fit the distance between *C* and *G*, as well as each of their distances from the user’s starting point—not just how they are related to the user, but how they are related to *each other*. If the user is at the pole and the spherical coordinates of the other points are $C=(\theta_{C},\varphi_{C})$, $G=(\theta_{G},\varphi_{G})$, the symmetries of the sphere (including global phase-invariance) ensure that the absolute phase values do not matter—the polar angles $\theta_{C}$ and $\theta_{G}$ and the difference in phase angle $\varphi_{GC}=\varphi_{C}-\varphi_{G}$ are enough to describe the whole scenario. This new parameter $\varphi_{GC}$ can be interpreted as the extent to which the user perceives the two other items or questions to be related to each other. (The notion that semantic vector similarity should take that into account a user’s point-of-view was also used by Aerts et al. [40].)

Thus, a single qubit has exactly the right number of coordinates/degrees of freedom to model the order effects in two-question scenarios such as the Clinton–Gore example. In general, it is more desirable to have fewer parameters than data degrees of freedom, since exact equivalence undermines the necessity that a model has a particular form. (However, we will see in subsequent sections that for larger circuits, a 1-1 correspondence between degrees of freedom in the circuit and parameters of a psychological model is not obvious in higher dimensions.) It also preserves an interesting symmetric property whereby the gain in trustworthiness for Clinton when compared with Gore is the same as the loss in trustworthiness for Gore when compared with Clinton. This mathematical property is described as ‘QQ-equality’ by Wang et al. [41], in whose survey analysis it is found to hold with surprising accuracy.

The three circuits used to model the scenario and recover the probabilities for the questions “Is Gore honest?” and “Is Clinton honest?”, with and without being asked the Gore question first, are shown in Figure 6. They are slightly more complicated than might be expected because, in practice, there are also constraints that arise from running on contemporary NISQ (Noisy Intermediate Scale Quantum). Some of these choices are described in detail by Nielsen and Chuang [37] (§4.4), including the *Principle of Deferred Measurement*.

Instead of measuring the state |0〉 in the basis of the *Clinton* or *Gore* axes, an appropriate rotation is applied and then the state is measured in the usual computational basis. These are mathematically equivalent, and enable measurements along any notional axis to be performed while only physically implementing measurement for one axis (typically the *z*-axis). So, to check “Is *A* honest?” with honest being the |0〉 state, we set the state to *A* and check for honesty, rather than setting the state to honesty and checking for *A*. Thus, instead of measuring along the *Gore* axis, we perform an *X*-rotation through angle $\theta_{G}$ and then measure along the standard *z*-axis to see if the qubit is in state |0〉 or |1〉. Or, if we choose not to measure this state but only the *Clinton* question without the comparative *Gore* context, we append the gate operators to map from the *Gore* state to the *Clinton* state and measure that instead. This gives the topmost circuit in Figure 6.

The other two circuits in Figure 6 model the situation where the question “Is Gore honest?” is asked first. However, measuring the same qubit more than once during a quantum circuit is error-prone. This is sometimes described as the *mid-circuit measurement* problem, and though the recent study of Hua et al. [42] “discovered non-trivial potential for qubit reuse”, this is early-stage and not yet used widely. In our implementation, instead of using one qubit and measuring its state twice, two qubits are used, which can be swapped with one another after the first $\theta_{G}$ rotation. Such a placeholder qubit is often called an *ancilla* (c.f. ‘ancillary’). The final state in the main qubit is the same as the state that would result whenever the state that is measured mid-circuit is the same as the initialized state of the qubit that was swapped in. Hence, the difference between the middle and the bottom circuit in Figure 6: the initial full *X*-rotation on the ancilla qubit makes it so that the |1〉 state is inserted instead of the |0〉 state, which corresponds to answering the first question “Is Gore honest?” with “No”.

There are various ways such a protocol can be implemented as well as swapping qubits: as an alternative solution to the same problem, Orrell [10] (§5.5) proposes the use of two qubits connected by a CNOT gate. Such techniques reduce the engineering problem from implementing many reliable gate options, and many reliable measurement options, to keeping the same requirements on gate implementation, while reducing the requirements for measuring reliably along any axis at any time.

Using these substitutions, the probabilities for what would have happened had the intermediate state been measured can be reassembled, but at the cost of using an extra qubit and circuit complexity. Thus, the second and third circuits in Figure 6 give the probability of saying that Clinton is honest and Gore is honest/dishonest, depending on whether the ancilla qubit q2 is *X*-rotated before being swapped for q1. The relative proportions of the relevant outcomes are then used to reconstruct the conditional probabilities as if the intermediate state was measured. For example, in the bottom circuit of Figure 6, the ancilla qubit $q_{1}$ is swapped with $q_{0}$ in the state representing the answer “Gore is dishonest”, and so the frequency of answering with the combination “Clinton is honest given that Gore is dishonest” is given by the conditional probability $P(q_{0}=0,q_{1}=1|q_{1}=1)$, i.e., the ratio $\frac{\#|01\rangle }{\#|01\rangle \;+\;\#|11\rangle }$.

This example is quite simple, but already demonstrates some of the issues and opportunities when implementing quantum models from the literature on quantum computers. The use of complex numbers and combinations of qubits give rise to opportunities to build models and fit parameters in new ways, and in particular, the extra phase parameter $\varphi_{GC}$ enables us to model the relationship between three states (the participant’s initial state, and the states representing the answers to two questions) exactly and without redundancy for this small model. The design choices made when implementing this model using quantum circuits also depend on current hardware features, some of which are evolving quite quickly.

### Conditions on Probabilities in Order Effect Circuits

Quantum probabilities are constrained by an axiomatic framework analogous to that of classical probability theory, but with notable differences. Some of these differences depend on choices of implementation for particular operators. For example, the belief “Clinton is honest, and Gore is honest” could be represented as an intersection of higher-dimensional subspaces representing “Clinton is honest” and “Gore is honest”, respectively. This is like the original quantum logic of Birkhoff and von Neumann [43]. However, if the beliefs “Clinton is honest” and “Gore is honest” are represented just as one-dimensional lines (equivalent to points on the Bloch sphere for an individual qubit), their intersection is typically empty or trivial (and even if these concepts are represented as higher-dimensional subspaces, the projection onto the intersection of these subspaces is only well-defined when the projections onto each subspace commute with one another [44], §9.2.3).

Instead, our order effects circuit models use the notion that the probability that a participant says “Clinton is honest” and “Gore is honest” is given by the projection onto the *Clinton* state followed by projection onto the *Gore* state. Projecting onto *A* and then onto *B* is sometimes called an ‘&then’ operator, following Busemeyer et al. [22] and Pothos and Busemeyer [2]. Conditions associated with this operator include:In quantum theory, we can have P(A&thenB)>P(B), because projecting onto *A* and then onto *B* can give a higher probability than projection directly onto *B*.However, it has to be the case that P(A&thenB)<P(A) (so this model does not allow double conjunction fallacies).Since P(A&thenB)≠P(B&thenA), exchanging the orders of *A* and *B* terms in these conjunction expressions can lead to different outcomes.

Large differences between P(A&thenB) and P(B&thenA) can occur. For example, if the initial state is |0〉 and the *A* state is |1〉, then P(A)=0, so also P(A&thenB)=0. If the *B* state is $\frac{1}{\sqrt{2}}\left(0+1\right)$, half-way inbetween, then the probability of projecting onto *B* is $\frac{1}{2}$, the probability of projecting from there onto *A* is also $\frac{1}{2}$, so the combined probability is $\frac{1}{4}$. This property is sometimes referred to as the *quarter law* [10] (§4.7). Other basic axioms and properties of quantum probability are described by Busemeyer et al. [45] and Yukalov and Sornette [46].

In classical logic and probability, these differences do not arise: asking “Is the system in state *A*?” does not affect the state of the system itself, and modeling a conjunction as an intersection of sets, or as two set membership questions asked in succession, amount to the same thing. It follows that any system where P(A&thenB)≠P(B&thenA) is not classical in this sense. This is just a restatement of the mathematics behind Heisenberg’s Uncertainty Principle: if two operators do not commute with each other, then measuring the output of one operator first affects the behavior of the second. (Since quantum measurements are inherently probabilistic, establishing empirically that a system is behaving non-classically requires several experiments or shots, and statistical analysis to show that differences are significant, which motivates concepts such as the *shot noise level* (SNL) of Ukai et al. [47].)

## 6. Extending the Order Effects Model with Subjective Bias Activation

This section describes a useful extension to the order effects circuits to include a selective behavior, so that the first question is asked only when a given initial condition is met for that participant.

Consider the circuit in Figure 7. It is just like the bottom circuit in Figure 6, except that there is an extra qubit $q_{2}$ which controls the swap gate. In other words, the swap gate (which simulates asking the intermediate question) only operates if $q_{2}$ starts in an excited |1〉 state, rather than a ground |0〉 state. This means that the circuit will behave differently based on the input state: so if the input state is a model for the participant’s initial state or point-of-view, this can be used to model what happens when users start from different states.

Note that as the circuits we consider have more qubits, there are more parameters that could be used but are not. For example, in the ancilla qubit $q_{1}$ and the new priming qubit $q_{2}$, we have only considered them as being activated or unactivated, without considering partial activation and whether the use of complex numbers introduces new opportunities for phase interference effects. In theory a state with *n* qubits can represent $\;2^{n}$ complex parameters (or more strictly, $2^{n}$− 2 because normalization and global phase-invariance remove 2 degrees of freedom), and as models grow, it is unlikely that we will have an individual role for each parameter. We regard the number of parameters as a computational feature rather than a psychological reality: by analogy, in classical computational modeling, the fact that bits are grouped into bytes of 8 bits does not mean that the number of parameters used in classical computational models should always be divisible by 8.

In general, $q_{2}$ can simulate attentional processes, which determine which questions are considered in a thought process—and so can lead to individual differences. A similar design was used by Kvam and Pleskac [48] to model the interaction between user beliefs and user judgments: in particular, the extra ‘belief’ qubit is excited when a user recognizes a concept and this cue can influence subsequent beliefs or decisions:

If an indeterminate cue influences beliefs (via a U-gate operation), evaluation of the cue should affect subsequent evaluation of beliefs and information about the criterion should affect beliefs about the cues [48].

This can be used to explain why different people respond differently to different cues. The argument can be put differently from the perspective of information overload leading to disagreement presented by Pothos et al. [49], which considers “well-meaning Alice and Bob debating a complex political question” and assumes they “share their questions and outcomes”. This shows that dysfunctional disagreement can still happen if the information is too complicated to consider without simplifying generalizations, and Alice and Bob start with incompatible simplification strategies. Instead, the model in Figure 7 assumes that Alice and Bob may have different states for qubit $q_{2}$, and this guides them concerning which intervening questions to ask: even if in theory Alice and Bob ‘see’ the same information, some information will be considered and some will be ignored, based upon Alice and Bob’s prior state. It is quite possible for this to lead to feedback whereby Alice and Bob continue to make different choices about which information they accept and process and which information they ignore: instead of information overload, the circuit in Figure 7 models something more like selection bias.

A particular example where this can be seen is in some news headlines: for example, it is more common in the USA for headlines to say that a court-case is “a clash between religious freedom and gay rights”, rather than (for example) “a clash between employment law and customer service guarantees” because the topics “religious freedom” and “gay rights” are more likely to engage readers than ‘employment law’ or ‘customer service guarantees’. Headlines are deliberately written to connect a story with something the user already cares about, which makes it more likely that the user will read the story.

The notion of conditional activation can be implemented classically, of course: for example, the representation for each user in the model could include a dictionary of configuration parameters, and the implementation could check these parameters and execute an explicit ‘if … then …’ clause that changes behavior based on these parameters. This approach is more like a user asking explicitly “Based upon my prior experiences, am I likely to be engaged by this content?” (which in a quantum circuit would correspond to an explicit measurement step). Part of the attraction of the alternative quantum model in Figure 7 is that these correlations and choices can be modeled implicitly, which allows for unexpected correlations and behaviors which could be characterized as more instinctive. (In artificial intelligence terms, the classical conditional implements a tiny rule-based expert-system, and the quantum circuit implements a tiny quantum neural network.) Additionally, in a quantum framework, asking intermediate questions has the potential to alter the underlying mental states, in a specific way, something which does not occur naturally in classical approaches [50].

Instead of humans being modeled as rational actors experiencing the same information events and trying to converge to similar beliefs, in this approach, a human’s state-of-mind may include a web of activated entangled concepts, which can even be manipulated to make us engage with new information in different ways. This description of what humans are like is very different from the ‘rational actors’ often preferred in scientific models since the enlightenment (mid-1600s); however, in the 2020s, there is widespread acknowledgment that social and personalized media have led to widely divergent self-reinforcing information and belief systems. Despite the discomfort, this day-to-day evidence has made it much easier for scientists to accept that, as humans, we are all biased and selective in our approach to information, and that this is something we must take into account rather than try to ignore.

## 7. Quantum Circuits for Disjunction Effects

Turning to our other main class of cognitive decision models, this section shows how a quantum cognitive model for disjunction effects (as described in Section 3.2 and [5] (Ch. 4), [10] (Ch. 4)) can be implemented using quantum circuits. Common themes of these models include:A method to set the probability of a single event.A method to connect events saying that the outcome of a particular event may make an output of a subsequent event more or less likely.A method to ‘entangle’ events so that states representing different potential events can interfere with one another, including interference between incompatible outcomes.A method to ‘measure’ events, to model what happens when we learn the outcome of one of the hitherto unknown events and remove the possibility of other outcomes.

This section demonstrates a combination of basic circuit elements that implement these four key processes.

The process for setting the probability of a single event is just as described above in Figure 4, so we proceed to explain the other three components.

### 7.1. Connecting Dependent Events

To implement a connection between events that represents the conditional probability P(B|A), we declare qubits $q_{A}$ and $q_{B}$ to represent each event. We then add a two-qubit gate that implements a partial rotation on the target qubit $q_{B}$, conditioned on the control qubit $q_{A}$. The angle of this partial rotation is given by the standard transformation from probabilities to angles, $\theta =\arccos(\sqrt{P(B|A}))$. The circuit in Figure 8 is an example that implements a basic combination of changing the probability of *event 2*, based on whether *event 1* happens or not. Similar conditional probability circuits are described by Borujeni et al. [51] in the development of quantum Bayesian networks.

These two components are enough to implement circuits that are equivalent to a classical Bayesian network connecting events 1 and 2, which looks like the circuit in Figure 9.

This circuit obeys the classical law of total probability, in this case, the rule that $P\left(B\right)=P\left(B|A\right)P\left(A\right)+P\left(B|A^{\prime }\right)P\left(A^{\prime }\right)$. The quantum circuit components introduced so far have been demonstrated to give the same outcomes as their classical counterparts. That this is possible with just 2 qubits is striking, but the circuits do not yet model violations of classical rules, such as those seen in disjunction effects.

### 7.2. Interference between Unknown Outcomes

The quantum circuit used to implement interference between unknown outcomes is based on the intuition behind a Mach–Zehnder Interferometer (Figure 10). In such a system, a half-mirror splits a beam into two parts, one of those parts may undergo a phase shift, and when the beams are brought back together, they interfere constructively or destructively, based on the phase angle.

A quantum circuit that behaves the same way is described by authors including Lee et al. [52] and shown in Figure 11. It uses Hadamard gates as the ’beam splitter’ that maps a qubit in the state |0〉 to the state $\frac{1}{\sqrt{2}}\left(|0\rangle +|1\rangle \right)$, though this is not the only option. For example, changing the single-qubit gates on the target (lower) qubit in Figure 12 changes the output probabilities to different ranges. Other operations can be used to shrink the window of variation from one-half to smaller ranges. In general, the group of possible two-qubit operations is isomorphic to U(4), the 16-dimensional group of 4×4 complex-valued matrices which conserve probabilities, so there are many more degrees of freedom available than just the φ parameter in Figure 11. On the one hand, this proliferation makes it harder to make direct correspondences between model parameters and psychological factors. On the other, it supports a rich variety of representations. More work is needed to understand which families of representations are appropriate for modeling behavioral findings.

This is not the only option. For example, just applying the gates $H\to R_{z}\left(\varphi \right)\to H$ on the same qubit will produce interference between the two states in a single qubit. Instead, using two-qubit configurations to generate interference enables the model to implement an intuition such as “the uncertainty in event 1 affects the phase change in event 2”. This leads to the circuit structure of Figure 11. Various gates could be used instead of the Hadamard (H) gates, which lead to different relationships between the phase angle φ and the probability for event 2. Some of these combinations are shown in Figure 12. The decision to use Hadamard gates is a choice that says “if the probability of event 1 is zero, the probability of event 2 will be between zero and one-half”. (This is before any explicit conditional probabilities, based on knowing the outcome of event 1, have been included.)

The three components so far are assembled into the following ‘conditional probability with interference’ circuit shown in Figure 13. Figure 14 shows how this circuit behaves with the numbers filled in from the Prisoner’s Dilemma problem above. Event 1 is the partner’s decision result, with state |0〉 corresponding to ‘Betray’ and state |1〉 corresponding to ‘Cooperate’. Event 2 is the participant’s decision, with the same correspondences.

The angles represent the average probabilities reported above by Busemeyer and Bruza [5] (Ch. 9). For the probability of event 1 itself (the partner defects), a value of 50% is commonly used, reflecting the fact that the participants are not given any prior estimate of this event.

The key finding is that the probability of event 2 (the participant defects) occurring varies with the phase angle φ, and it is possible to find a value of φ for which the estimated probability is the same as that observed in experiments.

### 7.3. Measuring an Outcome and Removing Interference

The three components assembled so far can generate the expected probability of event 2 if the outcome of event 1 is unknown. The remaining question is what happens when the outcome of event 1 (such as partner betrays/cooperates) becomes known. In quantum theory, this corresponds to measuring the qubit representing event 1, at which point it collapses to the pure |0〉 or |1〉 state. As in Section 5, this can sometimes be modeled without mid-circuit measurement using swap gates and ancilla qubits.

An example is shown in Figure 15. In this case, the *X* rotation on the “ancilla measure event 1” qubit is used to simulate measuring a |1〉 rather than a |0〉 for the first event. Swapping in a |0〉 qubit for event 2 corresponds just to resetting this qubit.

This component is added between the interference component and the conditional probability component discussed above, giving the final Prisoner’s Dilemma circuit in Figure 16. This circuit recreates all the desired outcomes for the Prisoner’s Dilemma problem. The same circuit structure with different parameter values can be shown to recreate other well-known disjunction problems, including the Two-Stage Gamble and Hawaii Vacation problems studied since their introduction by Tversky and Shafir [15].

While each of the four circuit components used here can be found in different parts of the quantum information literature, we believe that this combination of components is novel, that it accurately models disjunction interference effects from cognitive science, and, to the best of our knowledge, it is the first quantum circuit to achieve this.

## 8. Developing and Running on Quantum Hardware

Since all of the circuits in this paper use 4 qubits or fewer, examples of the circuits from Figure 6 for order effects and Figure 16 for disjunction effects could be run easily in the regular duty cycle of an 11 qubit machine made and maintained by IonQ [6]. The qubits in this machine are made of Ytterbium ions (${}^{171}{\mathrm{Yb}}^{+}$), whose ground and excited energy states are distinguished by the hyperfine structure interactions between the spin of the nucleus and the angular momentum of the atom [53] (Ch. V). The single-qubit and two-qubit gates are controlled by Raman laser pulses, and the accuracy of these operations is reported as 99.5% and 97.5%, respectively. Each circuit was run with 10,000 shots (repetitions), which is a key consideration in quantum computing because the output is classical measurement results, not internal quantum states. These measurements are probabilistic by nature, so normally a non-trivial sample collected over many different experiments or shots is necessary.

The circuits for the Prisoner’s Dilemma (three versions: partner betrays, partner cooperates, partner decision unknown) produced the expected probabilities (82%, 73%, 64%) very accurately, with the largest discrepancy being just 0.37%. The circuits for the Clinton–Gore order showed more variation: the probability of predicting “Clinton is honest” with and without being asked the same question about Gore tended to be within 2–3% of their expected values. This is close to the reported two-qubit gate fidelity for the circuits with swap gates, so circuit-specific sources of error were not investigated separately.

These results do not demonstrate a ‘quantum advantage’ in the sense of running something that would be intractable on a classical computer. Instead, they demonstrate that we can start with the intended behavior of a small psychological model, and implement small quantum circuits that demonstrate key statistical outcomes of these models. The present results have some immediate interest, partly because it is sometimes surprising how much can be represented with just a few qubits and gates. (By contrast, a circuit using classical bits would require several times more bits and polynomially more gate operations, but these are so much simpler and easier to build using current technologies that a detailed algorithmic comparison would be actively misleading.)

More exciting opportunities here will arise as these components are connected into larger networks. Rather than making the initial experiments here obsolete, this could heighten the value of small software components used to make complicated quantum circuits, because programming and testing can be conducted by arranging these higher-level components, rather than addressing individual qubits and gates.

## 9. Related and Further Work

This paper has combined work in quantum circuits and quantum cognition. Several of the quantum circuit ingredients here have been used before: for example, the quantum circuit used for interference effects (Figure 11) was introduced as a general ‘quantum Rosetta Stone’ by Lee et al. [52]. It makes sense to try and assemble larger quantum circuits out of such components to produce higher-level structures such as the belief-activation circuits of Kvam and Pleskac [48] and the quantum Bayesian networks of Borujeni et al. [51]. The availability of real quantum hardware has encouraged such models to be run on quantum computers in related areas, such as natural language processing [8,9], and it is natural to expect similar progress in quantum cognition over the next few years.

Quantum cognition has itself been developing the use of corresponding quantum models for many years [22,54]. The use of complex numbers and the exponential growth in available parameters give more options for parameter fitting, and more challenges in interpreting and using these parameters, a situation previously studied by Moreira and Wichert [24] with heuristic approaches to parameter selection.

Further work should include adapting these techniques to larger systems and datasets. Even the small components discussed here could be rearranged in several ways: for example, the output of a phase interference kernel (Figure 11) could be used as the input state that may or may not activate a subjective bias entanglement (Figure 7).

It is possible that such properties could be employed for cognitive findings which have previously benefited from quantum models using entanglement, such as conceptual combination [55,56] or bipartite games, such as Prisoner’s Dilemma, with communication [57]. A particularly interesting direction for application is whether quantum circuits can offer new perspectives on more efficient memory architectures, taking advantage of superpositions and entanglement to multiply query memory representations.

## 10. Conclusions

Quantum cognition is a well-established theoretical field that has demonstrated good overlap with real world situations, qualitatively and quantitatively, that violate constraints that would apply if the decision options are considered separately and their likelihoods inferred using classical probability laws. The mathematics of these models (particularly linear algebra) is used independently of quantum mechanics, just as the mathematics developed for classical mechanics (particularly calculus) has long been used in social sciences. Before 2020, this research program was developed largely independently of quantum computing, but we are now able to run some of these models on quantum computers for the first time.

Implementing such models on quantum computers is often not straightforward, partly because the approach to computing is unfamiliar to most programmers, partly because sometimes alternatives need to be chosen based on a knowledge of current hardware support and limitations. One of the main methodological differences between classical and quantum computing is that even a single quantum bit (qubit) provides a much richer structure than a classical bit and, as qubits are added, this richness grows exponentially.

Quantum programming provides an opportunity to consider new approaches to programming and modeling. In classical computing, at least in theory, all states are known and all operations are predictable. In quantum computing, a state can be unknown and is sometimes deliberately put into a probabilistic state, and deliberately not measured or scrutinized while it interacts with the rest of the system. This encourages different ways of thinking about state and configuration that are intriguing and unfamiliar: instead of listing rules to follow or functions to execute, quantum programming is more like preparing a collection of states and correlations, letting them evolve, and then probing some part of the eventual state with prepared ‘yes–no’ questions. Writing a quantum program as a design process has a lot more in common with designing a complicated psychological survey than classical programming. This is not in itself evidence that human thought is a kind of quantum computation; however, it may encourage quantum programmers to think about computation not only as a rigid set of steps to follow, but also as a way of preparing systems with a network of complicated correlations, which sometimes combine to give surprising results.

This paper has provided some initial quantum implementations, showing some ways in which quantum computers lend themselves naturally to these tasks and revealing some areas where hardware constraints still constrain design choices. We hope that more research in this area will lead to the establishment of common design patterns that are known to represent some of these non-classical aspects of human cognition naturally and accurately.

## Figures and Tables

**Figure 1 entropy-25-00548-f001:**
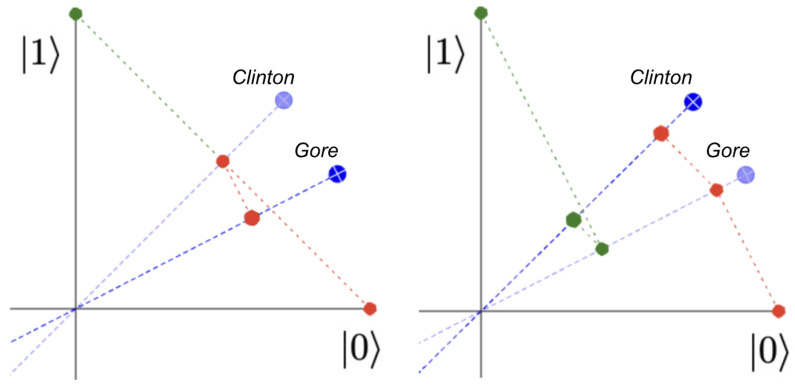
Non-commutative projections model the order effect in the Clinton–Gore scenario. We see that the projection of the |0〉 vector onto the *Clinton* axis gives a point further from the origin if we first project onto the *Gore* axis (**right**) rather than if we just project the |0〉 vector onto the *Clinton* axis (**left**). These diagrams show only the quadrant with positive real coordinates, so if the *Gore* axis is at angle θ above the horizontal, it appears at the point cos(θ)|0〉+sin(θ)|1〉. In reality, the coordinates can be any complex numbers α and β such that $|\alpha^{2}\left|+\right|\beta^{2}|=1$.

**Figure 2 entropy-25-00548-f002:**
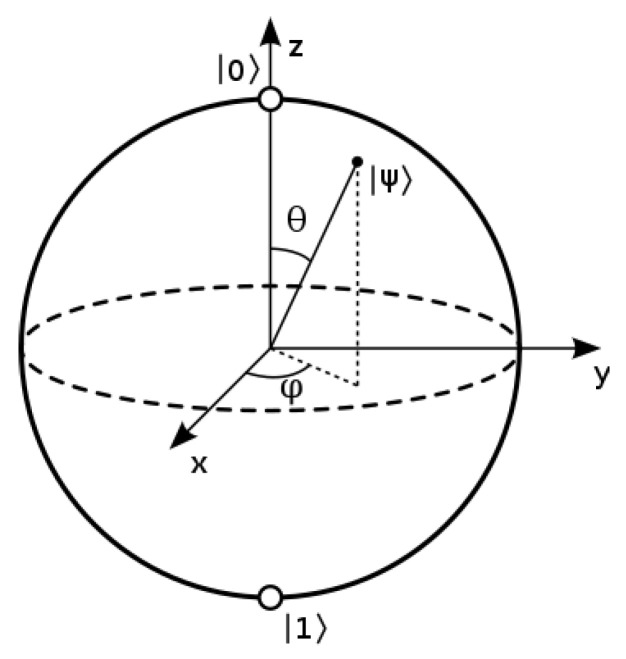
Bloch sphere representation of a qubit. (From https://en.wikipedia.org/wiki/Bloch_sphere, Creative Commons CC BY-SA 3.0 license. Accessed on 14 March 2023).

**Figure 3 entropy-25-00548-f003:**
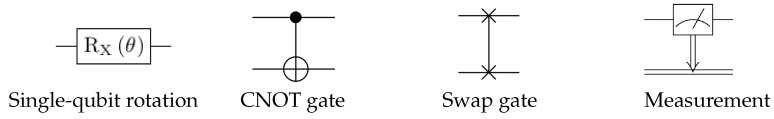
Basic quantum logic gate diagrams used throughout these examples. A single-qubit rotation gate manipulates the superposition of |0〉 and |1〉 states for the qubit. The two-qubit CNOT gate (right) entangles two qubits (the top qubit is the control qubit and the bottom is the target qubit). The swap gate swaps the states of the two qubits. The measurement operator measures the qubit’s value and stores it in the given classical bit.

**Figure 4 entropy-25-00548-f004:**
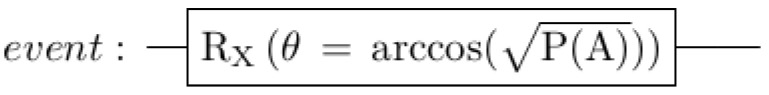
Circuit for setting probability of single event *A*.

**Figure 5 entropy-25-00548-f005:**
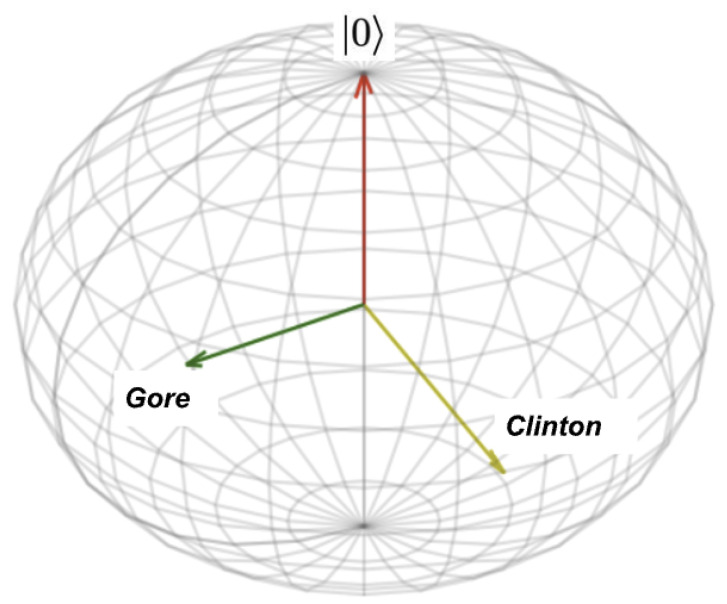
Bloch sphere vectors for Clinton and Gore.

**Figure 6 entropy-25-00548-f006:**
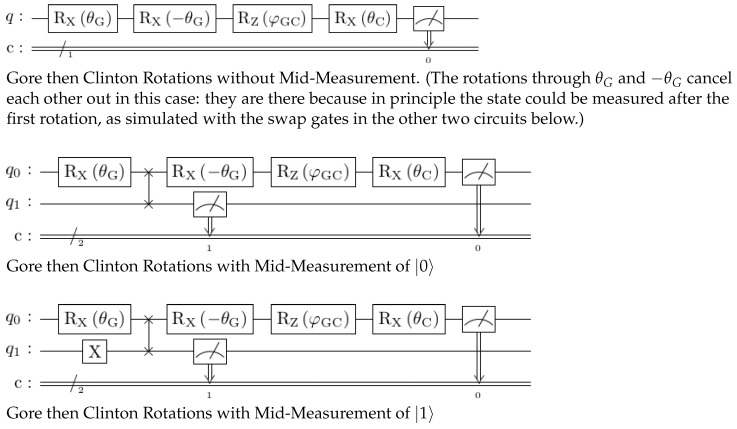
Circuits implementing the Clinton–Gore order effects, with and without mid-measurement.

**Figure 7 entropy-25-00548-f007:**
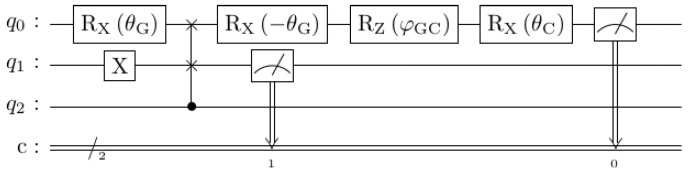
Order effect circuit with an extra qubit $q_{2}$ that controls whether or not the participant is asked the first question.

**Figure 8 entropy-25-00548-f008:**
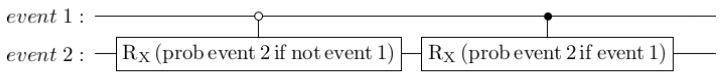
Circuit for setting conditional probability. Note that the white circle means ‘if this qubit is in state |0〉’ and the black circle means ’if this qubit is in state |1〉’.

**Figure 9 entropy-25-00548-f009:**
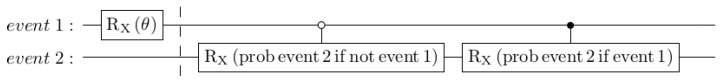
Circuit implementing a simple classical Bayesian network.

**Figure 10 entropy-25-00548-f010:**
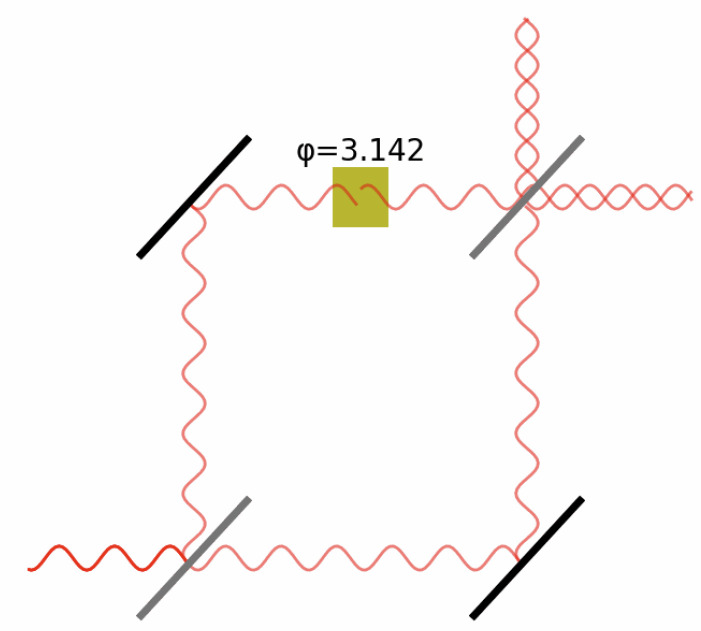
Schematic diagram of a Mach–Zehnder Interferometer.

**Figure 11 entropy-25-00548-f011:**
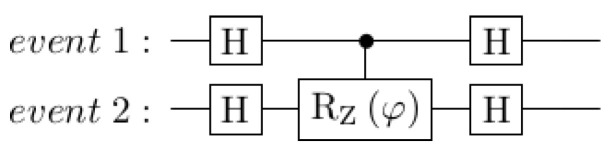
Circuit for simulating interference between unknown outcomes. The ‘H’ gate is a Hadamard gate which maps the state |0〉 to a superposition state $\frac{1}{\sqrt{2}}\left(|0\rangle +|1\rangle \right)$, and |1〉 to the state $\frac{1}{\sqrt{2}}\left(|0\rangle -|1\rangle \right)$.

**Figure 12 entropy-25-00548-f012:**
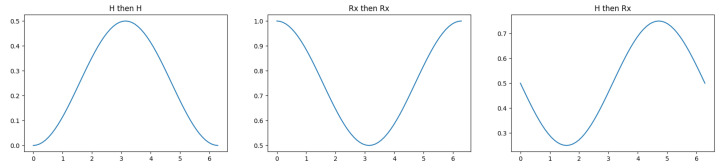
Different output probabilities for the target qubit as a function of the phase angle φ, when the gates before and after the $R_{z}\left(\varphi \right)$ operation of Figure 11 are changed.

**Figure 13 entropy-25-00548-f013:**
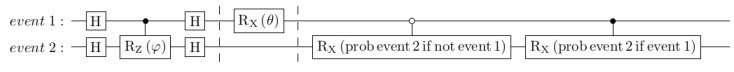
Circuit for conditional probability with interference.

**Figure 14 entropy-25-00548-f014:**
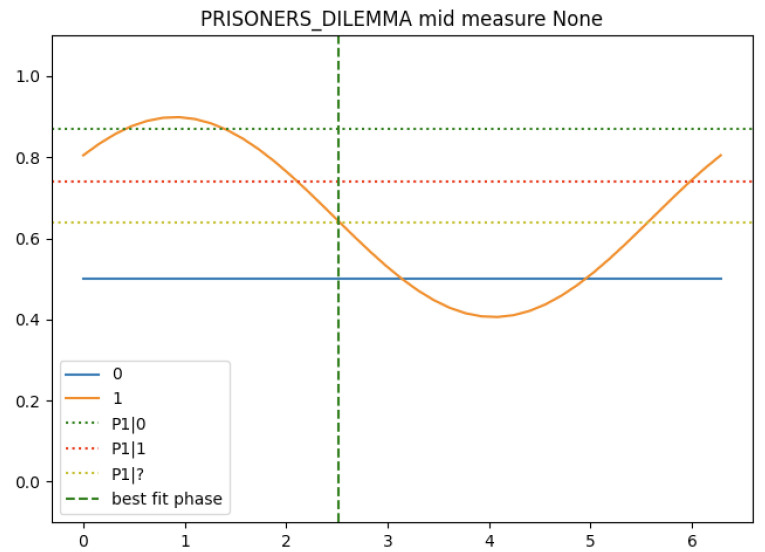
Outcome probabilities for the Prisoner’s Dilemma scenario.

**Figure 15 entropy-25-00548-f015:**
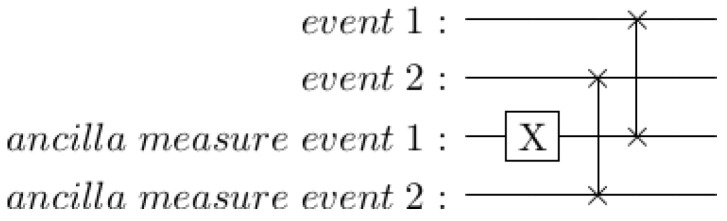
Swapping in ancillas as a proxy for mid-circuit measurement.

**Figure 16 entropy-25-00548-f016:**
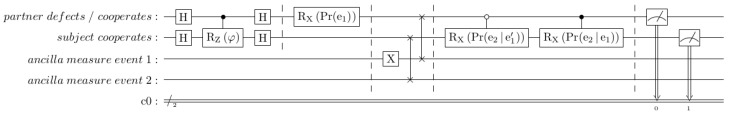
Complete circuit simulating the Prisoner’s Dilemma scenario.

## Data Availability

The data used in this paper is all from results previously published in other books and papers.

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
