# Peer review of "Quantum Circuit Components for Cognitive Decision-Making"

_entropy, 2023, doi:10.3390/e25040548_

Round 1
Reviewer 1 Report
The article shows some initial quantum circuit implementations that tie the cognitive and mathematical modelling together into intuitive building-blocks. It is novel to explain the concept of cognitive decision-making using a quantum circuit model, and the logic is relatively clear. Although it is puzzling in physics, the quantum circuits for order effects are clearly expounded. I can recommend it for publication in Entropy if the following points will be properly addressed:
1. It will be better to add description about differences and common characteristics between the quantum circuits in quantum computation and quantum circuits for order effects.
2. In quantum circuits for quantum computation, if the two levels of the system are written |0⟩ and |1⟩, a qubit can be in a more general state α|0⟩ + β|1⟩, then it follows |α2| + |β2| = 1. So what are the limiting conditions of the probability in quantum circuits for order effects?
3. In section 3.1, the authors indicate that “The initial |0⟩ state can be written as a superposition of “Clinton is honest” and “Clinton is not honest”.” In section 5, the authors define qubits as a superposition of “X is honest” and “X is not honest”. Are these two definitions contradictory? What is the physical meaning of the abscissa in Figure 1? It will be better to write the expression of qubits, rather than simply describing them.
4. As stated in ref. [Phys. Rev. Lett. 95, 170501 (2005), Phys. Rev. Lett. 106, 240504 (2011)], quantum circuits in quantum computation have a classical measurement boundary limit to distinguish the classical and quantum protocols. Then, in quantum circuits for order effects, a similar boundary condition should be put forward even if the results here do not claim to show ‘quantum advantage’. Or the authors can explain the other quantum characteristics of the model.
5. In section “7.2 Interference Between Unknown Outcomes”, the third paragraph is not complete.
6. The description of Figure 12 is missing in the manuscript.
Author Response
Thank you for the review comments and suggestions. Here is how we have addressed each one.
- It will be better to add description about differences and common characteristics between the quantum circuits in quantum computation and quantum circuits for order effects.
At the end of Section 4, we’ve added:
The quantum circuits that will be introduced in the next sections, for implementing quantum cognitive models including order and disjunction effects, use standard quantum computing circuit-construction methods: there are no new gates or operations needed that make quantum cognition circuits different from quantum computing circuits in general. However, there are limitations in the current generation of quantum computers (including the lack of mid-circuit measurement), which make some implementation choices more convenient than others today. These decisions are discussed in subsequent sections, as the situations arise.
- In quantum circuits for quantum computation, if the two levels of the system are written |0⟩ and |1⟩, a qubit can be in a more general state α|0⟩ + β|1⟩, then it follows |α2| + |β2| = 1. So what are the limiting conditions of the probability in quantum circuits for order effects?
We have added a section 5.1., Conditions on Probabilities in Order Effect Circuits, listing several such conditions. This includes a short discussion on order effects and non-commutativity (which also helps to cover point 4 below).
- In section 3.1, the authors indicate that “The initial |0⟩ state can be written as a superposition of “Clinton is honest” and “Clinton is not honest”.” In section 5, the authors define qubits as a superposition of “X is honest” and “X is not honest”. Are these two definitions contradictory?
Thanks for this point, but no, they are not contradictory. We assume we have a two dimensional basis for honesty, that is, we assume that we can make judgments concerning whether a person is honest or not honest. Then, we can specify a superposition state for any person (whether Clinton or X), in terms of the amplitudes (weights) for honesty vs. not honesty. That is, for one person, Clinton, we can write:
|Clinton> = a |honesty> + b |~honesty>, where |Clinton> is the mental state corresponding to Clinton and a, b the amplitudes for the two possibilities. For Clinton, we might guess that for most people |a|<|b|.
For another person, X, we might write
|X> = a’ |honesty> + b’ |~honesty>
where now we are using different weights.
In the manuscript, we’ve added a footnote in Section 5 saying:
This generalizes the point in Section 3 that the |0⟩ state can be written as a superposition of states representing “Clinton is honest” and “Clinton is dishonest”. The |0⟩ state can be written as such a sum for any pair of orthogonal vectors X and X′, and any such X can be written as a superposition of |0⟩ and |0⟩′=|1⟩.
What is the physical meaning of the abscissa in Figure 1? It will be better to write the expression of qubits, rather than simply describing them.
Added more explanation to the caption to Figure 1, saying:
These diagrams show only the quadrant with positive real coordinates, so if the Gore axis is at angle θ above the horizontal, it appears at the point cos(θ)|0⟩+ sin(θ)|1⟩. In reality the coordinates can be any complex numbers α and β such that |α2|+|β2|=1.
- As stated in ref. [Phys. Rev. Lett. 95, 170501 (2005), Phys. Rev. Lett. 106, 240504 (2011)], quantum circuits in quantum computation have a classical measurement boundary limit to distinguish the classical and quantum protocols. Then, in quantum circuits for order effects, a similar boundary condition should be put forward even if the results here do not claim to show ‘quantum advantage’. Or the authors can explain the other quantum characteristics of the model.
The key difference is that quantum models can demonstrate order effects (different results depending on asking questions in different orders), which is based on a non-commutative property that is absent in classical models. This has been added at the end of the new section 5.1 which introduces constraints and limits on classical and quantum probability.
- In section “7.2 Interference Between Unknown Outcomes”, the third paragraph is not complete.
Thanks. After “and the decision to use Hadamard (H) gates” we added
is a choice that says ``if the probability of event 1 is zero, the probability of event 2 will be between zero and one-half''. (This is before any explicit conditional probabilities, based on knowing the outcome of event 1, have been included.)
- The description of Figure 12 is missing in the manuscript.
Thanks, we added:
Various gates could be used instead of the Hadamard (H) gates, which lead to different relationships between the phase angle Φ and the probability for event 2. Some of these combinations are shown in Figure 12.
Reviewer 2 Report
Broadly speaking, the article comes across to me as a scholarly contribution and one that is well communicated. From the perspective of the psychological sciences I am able to offer a few comments and suggestions.
1. As a non-expert I think it would be helpful to make more clear (note: fully respecting the effort that has already been made by the authors) in what ways quantum cognition and quantum computing are alike, are distinct, and the ways in which they are known to or potentially could relate. Specifically, what are the implications of bringing the two quantums together? Why is this specifically important/challenging and how can the authors contribution be understood in these terms? The leap to the second half of the article is perhaps unavoidably going to be tough sledding for the uninitiated.
2. Other formal/mathematical approaches (e.g. connectionism, bayesian) in cognitive science have succeeded in part by providing a 'theoretical vocabulary' for researchers, i.e., a set of constructs, elements, design principles that enhance the potential for formulating clear, novel explanations and predictions in the field (also that can be expressed in a level above the fully technical specification). My point is: I could not really see something like this coming across in this manuscript, but I think it would be very helpful if it did.
3. The discussion of biased cognition, departures from rationality, and classical logic is effective. However, I would point out that nobody I talk to really takes seriously the notion that human cognition is effectively characterized by anything like a classical logic-based system. So I think it would be a big improvement to the paper if the quantum view was framed and evaluated -- not in contrast with classical logic -- but with other alternative explanatory views that have been put forward in cognitive science (off the top of my head: dynamic systems, statistical learning, evolutionary psychology, connectionism, socio-motivational theory, social cognition, ...). The point is that psychologists have some promising explanations for why various empirically-supported biased / irrational phenomena occur and I would like to see the quantum approach put forward with respect to such contributions (rather than as seemingly the sole alternative to classical logic). To dig a little further on this, a perhaps compelling perspective is that people are often not answering the question they are given (particularly when facing uncertainty) but instead answering a question they prefer to deal with or taking an approach that satisfies goals other than accuracy (i.e., closure, satisfaction, convenience, justifiability, signaling). Does explanation of this kind run in contrast with the quantum approach or is it compatible (or something else)?
Author Response
Thank you for the review and suggestions. We address them below, and have used points 1 and 3 to make the following additions to the manuscript. Please consider the response to point 2 particularly, and let us know if some of the material there (particularly the list of familiar structures from quantum information processing) should be added to the manuscript.
- As a non-expert I think it would be helpful to make more clear (note: fully respecting the effort that has already been made by the authors) in what ways quantum cognition and quantum computing are alike, are distinct, and the ways in which they are known to or potentially could relate. Specifically, what are the implications of bringing the two quantums together? Why is this specifically important/challenging and how can the authors contribution be understood in these terms? The leap to the second half of the article is perhaps unavoidably going to be tough sledding for the uninitiated.
Thanks for this important comment. We have added a Section 4.1, Why Bring Quantum Cognition and Quantum Computing Together?, to discuss this point.
Both quantum cognition and quantum computing are constrained by the computational/ probability rules in quantum theory. Existing quantum cognitive models are intended to be at the ‘computational level’ of explanation, using Marr’s (1982) framework. The computational level concerns the what and the why questions for the system that is studied, that is “what is the goal of the computation, why is it appropriate, and what is the logic of the strategy by which it can be carried out?” (Marr, 1982, p.25). Notably, quantum cognitive models address the question of the computational principles which appear to guide behavior.
Quantum computing/ circuits offer insight into how quantum cognitive models could be implemented. In Marr’s terms, they concern the algorithmic level of explanation, which concerns process explanations of the studied system, specifically the representations that are employed by the system and the algorithms that operate on the representations to produce an output from an input.
With these thoughts in mind, a quantum circuits implementation of a quantum cognitive model has two main possible aims. First, it could serve as an algorithmic/ process proposal for a corresponding quantum cognitive model. Assuming that there are no real quantum mechanical processes in the brain, a putative brain quantum circuit would be epiphenomenal. Second, a quantum vs. classical circuit implementation (of a quantum cognitive model) bears on the question of human vs. artificial intelligence capacity and limits.
Both these aims would require extensive further work to substantiate – the present work provides the foundation for such work, by offering the first principled proposal for quantum circuits corresponding to a quantum cognitive model and proof of concept, in the form of the simulations conducted directly on a quantum computer.
A related reason for applying quantum computers to quantum cognition is the obvious motivation-by-opportunity --- ``because we can’’. A great deal of investment has gone into developing quantum computers, with anticipated application areas including materials science (e.g. molecular simulation) and logistics (because of the combinatoric complexity of such problems). It is to be expected that researchers will try to apply these machines to other areas, just as GPU's (Graphics Processing Units) have found extensive applications in machine learning, as well as computer graphics. We expect that, during the 2020's, quantum computers will be tried in many more application areas. Domains with established techniques that already use quantum mathematical models are naturally a promising place to start.
Marr, D. (1982). Vision: a computational investigation into the human representation and processing of visual information. San Francisco: W. H. Freeman.
- Other formal/mathematical approaches (e.g. connectionism, bayesian) in cognitive science have succeeded in part by providing a 'theoretical vocabulary' for researchers, i.e., a set of constructs, elements, design principles that enhance the potential for formulating clear, novel explanations and predictions in the field (also that can be expressed in a level above the fully technical specification). My point is: I could not really see something like this coming across in this manuscript, but I think it would be very helpful if it did.
You are absolutely right that more work is needed in this section: we tried hard to introduce (in a tutorial-like way) some of the key concepts. That is, part of the purpose of the paper is exactly to introduce cognitive scientists to the range of concepts technical tools in quantum circuits and how these could be employed in cognitive modeling. Examples include:
- Individual qubits, what they can represent, and in particular, the importance of complex numbers.
- The Bloch sphere representation.
- Basic quantum circuit gates, such as X rotations, CNOT, swap gates.
- The importance of measurement, and how it fixes quantum states.
- Quantum circuit concepts, such as entanglement, the use of ancillary qubits, and the principle of deferred measurement.
- The significance and challenges of exponential memory growth.
We could further emphasize any of these points, and / or could write such a list as part of the summary / conclusion. For now, we have decided against doing so, to keep the paper briefer and more to the point. However, this is easily rectified, if you think any of these additions would help.
- The discussion of biased cognition, departures from rationality, and classical logic is effective. However, I would point out that nobody I talk to really takes seriously the notion that human cognition is effectively characterized by anything like a classical logic-based system. So I think it would be a big improvement to the paper if the quantum view was framed and evaluated -- not in contrast with classical logic -- but with other alternative explanatory views that have been put forward in cognitive science (off the top of my head: dynamic systems, statistical learning, evolutionary psychology, connectionism, socio-motivational theory, social cognition, ...). The point is that psychologists have some promising explanations for why various empirically-supported biased / irrational phenomena occur and I would like to see the quantum approach put forward with respect to such contributions (rather than as seemingly the sole alternative to classical logic). To dig a little further on this, a perhaps compelling perspective is that people are often not answering the question they are given (particularly when facing uncertainty) but instead answering a question they prefer to deal with or taking an approach that satisfies goals other than accuracy (i.e., closure, satisfaction, convenience, justifiability, signaling). Does explanation of this kind run in contrast with the quantum approach or is it compatible (or something else)?
Thanks for this excellent point. This is a huge discussion, but in brief, we think it is reasonable to say the following (now Section 3.3: Bayesian Theory and the Cognitive Relevance of Classical Logic).
3.3. Bayesian Theory and the Cognitive Relevance of Classical Logic
The departure from classical logic and probability in this paper is not novel. Classical logic as such is no longer considered a viable approach to human behavior, and this was demonstrated convincingly, before quantum cognition started to take shape in the 1990s.
It has been generally accepted since the 1960s that human judgements violate classical logical rules, notably since the work of Wason [24]. Wason’s experiments on reasoning demonstrated systematic discrepancies, between conclusions drawn by participants, and
the “correct” inference, as predicted by the classical “if P then Q” material implication. Psychologists have explored a wide variety of frameworks for alternative theory and models, including heuristics and biases [25], and neural networks (as an example application, see Kurtz [26]).
However, for probabilistic reasoning specifically, an influential approach has been Bayesian probability theory. While it is easily recognized that baseline Bayesian theory cannot accommodate the range of relevant behavioral findings, researchers have sought bounded-rational versions of Bayesian theory [27,28]. The critical point is that the algebraic structure of Bayesian theory is exactly that of classical logic. Indeed, many of the apparent fallacies in probabilistic reasoning are so surprising, because they break seemingly obvious logical constraints (e.g., the conjunction fallacy, as explained in [2]).
While classical logic is no longer employed directly in cognitive modeling, its relevance is still current, because of the enduring interest in Bayesian cognitive models. An analogous point applies to classical circuits: classical circuits acquire ‘relevance’ in current theoretical discussions, exactly because they are the most direct way to implement Bayesian cognitive models. The work in this paper suggests that quantum circuits can play a similarly useful role in the implementation of quantum cognitive models.
[27] Costello, F. and Watts, P. (2014). Surprisingly rational: Probability theory plus noise explains biases in judgment. Psychological Review, 121, 463-480.
[25] Kahneman, D. (2001). Thinking fast and slow. Penguin: London, UK.
[26] Kurtz, K. J. (2007). The divergent autoencoder (DIVA) model of category learning. Psychonomic bulletin & review 14 (4), 560-576.
[2] Pothos, E. M. & Busemeyer, J. M. (2022). Quantum cognition. Annual Review of Psychology, 73, 749-778.
[24] Wason, P.C. (1968). Reasoning about a rule. Quarterly Journal of Experimental Psychology. 20, 273–281. https://doi.org/10.1080/14640746808400161.
[28] Zhu, J., Sanborn, A. N., & Chater, N. (2020). The Bayesian Sampler: generic Bayesian inference causes incoherence in human probability judgments. Psychological Review, 127, 719-746.
Round 2
Reviewer 2 Report
I'm satisfied with the authors responsiveness to comments.